# Evaluating Long-Term Memory in 3D Mazes

**Jurgis Pasukonis** *

DeepMind

Verses Research Lab

**Timothy Lillicrap**

DeepMind

University College London

**Danijar Hafner**

DeepMind

University of Toronto

## Abstract

Intelligent agents need to remember salient information to reason in partially-observed environments. For example, agents with a first-person view should remember the positions of relevant objects even if they go out of view. Similarly, to effectively navigate through rooms agents need to remember the floor plan of how rooms are connected. However, most benchmark tasks in reinforcement learning do not test long-term memory in agents, slowing down progress in this important research direction. In this paper, we introduce the Memory Maze, a 3D domain of randomized mazes specifically designed for evaluating long-term memory in agents. Unlike existing benchmarks, Memory Maze measures long-term memory separate from confounding agent abilities and requires the agent to localize itself by integrating information over time. With Memory Maze, we propose an online reinforcement learning benchmark, a diverse offline dataset, and an offline probing evaluation. Recording a human player establishes a strong baseline and verifies the need to build up and retain memories, which is reflected in their gradually increasing rewards within each episode. We find that current algorithms benefit from training with truncated backpropagation through time and succeed on small mazes, but fall short of human performance on the large mazes, leaving room for future algorithmic designs to be evaluated on the Memory Maze. Videos are available on the website: https://github.com/jurgisp/memory-maze

## 1 Introduction

Deep reinforcement learning (RL) has made tremendous progress in recent years, outperforming humans on Atari games (Mnih et al., 2015; Badia et al., 2020), board games (Silver et al., 2016; Schrittwieser et al., 2019), and advances in robot learning (Akkaya et al., 2019; Wu et al., 2022). Much of this progress has been driven by the availability of challenging benchmarks that are easy to use and allow for standardized comparison (Bellemare et al., 2013; Tassa et al., 2018; Cobbe et al., 2020). What is more, the RL algorithms developed on these benchmarks are often general enough

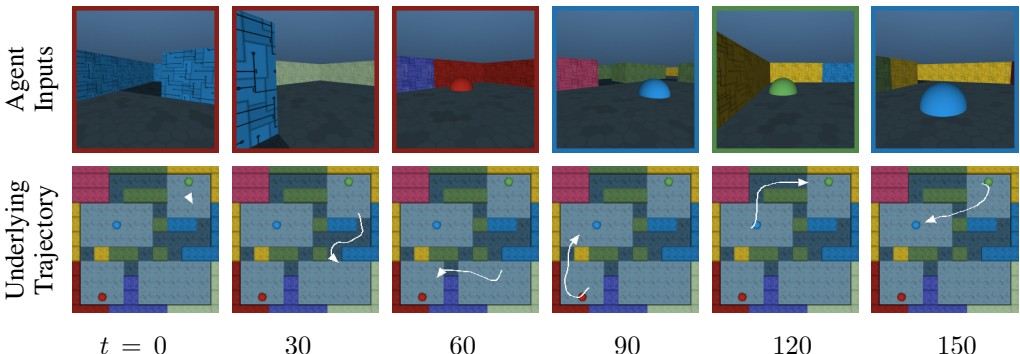

$t = 0 \qquad 30 \qquad 60 \qquad 90 \qquad 120 \qquad 150$

**Figure 1:** The first 150 time steps of an episode in the Memory Maze 9x9 environment. The bottom row shows the top-down view of a randomly generated maze with 3 colored objects. The agent only observes the first-person view (top row) which includes a prompt for the next object to find as a border of the corresponding color. The agent receives +1 reward when it reaches the object of the prompted color. During the episode, the agent has to visit the same objects multiple times, testing its ability to memorize their positions, the way the rooms are connected, and its own location.

---

* Corresponding author: Jurgis Pasukonis <jurgisp@gmail.com>

to solve completely unrelated challenges, such as finetuning large language models from human preferences (Ziegler et al., 2019), optimizing video compression parameters (Mandhane et al., 2022), or promising results in controlling the plasma of nuclear fusion reactors (Degrave et al., 2022).

Despite the progress in RL, many current algorithms are still limited to environments that are mostly fully observed and struggle in partially-observed scenarios where the agent needs to integrate and retain information over many time steps (Wayne et al., 2018). Despite this, the ability to remember over long time horizons is a central aspect of human intelligence and a major limitation on the applicability of current algorithms. While many existing benchmarks are partially observable to some extent, memory is rarely the limiting factor of agent performance (Oh et al., 2015; Cobbe et al., 2020; Beattie et al., 2016; Hafner, 2021). Instead, these benchmarks evaluate a wide range of skills at once, making it challenging to measure improvements in an agent's ability to remember.

Ideally, we would like a memory benchmark to fulfill the following requirements: (1) isolate the challenge of long-term memory from confounding challenges such as exploration and credit assignment, so that performance improvements can be attributed to better memory. (2) The tasks should challenge an average human player but be solvable for them, giving an estimate of how far current algorithms are away from human memory abilities. (3) The task requires remembering multiple pieces of information rather than a single bit or position, e.g. whether to go left or right at the end of a long corridor. (4) The benchmark should be open source and easy to use.

We introduce the Memory Maze, a benchmark platform for evaluating long-term memory in RL agents and sequence models. The Memory Maze features randomized 3D mazes in which the agent is tasked with repeatedly navigating to one of the multiple objects. To find the objects quickly, the agent has to remember their locations, the wall layout of the maze, as well as its own location. The contributions of this paper are summarized as follows:

- **Environment**   We introduce the Memory Maze environment, which is specifically designed to measure memory isolated from other challenges and overcomes the limitations of existing benchmarks. We open source the environment and make it easy to install and use.
- **Human Performance**   We record the performance of a human player and find that the benchmark is challenging but solvable for them. This offers an estimate of how far current algorithms are from the memory ability of a human.
- **Memory Challenge**   We confirm that memory is indeed the leading challenge in this benchmark, by observing that the rewards of the human player increases within each episode, as well as by finding strong improvements of training agents with truncated backpropagation through time.
- **Offline Dataset**   We collect a diverse offline dataset that includes semantic information, such as the top-down view, object positions, and the wall layout. This enables offline RL as well as evaluating representations through probing of both task-specific and task-agnostic information.
- **Baseline Scores**   We benchmark a strong model-free and model-based agent on the four sizes of the Memory Maze and find that they make progress on the smaller mazes but lag far behind human performance on the larger mazes, showing that the benchmark is of appropriate difficulty.

## 2   RELATED WORK

Several benchmarks for measuring memory abilities have been proposed. This section summarizes important examples and discusses the limitations that motivated the design of the Memory Maze.

**DMLab**   (Beattie et al., 2016) features various tasks, some of which require memory among other challenges. Parisotto et al. (2020) identified a subset of 8 DMLab tasks relating to memory but these tasks have largely been solved by R2D2 and IMPALA (see Figure 11 in Kapturowski et al. (2018)). Moreover, DMLab features a skyline in the background that makes it trivial for the agent to localize itself, so the agent does not need to remember its location in the maze. (Wayne et al., 2018) used a larger battery of tasks, but only a subset of those was included in the public release of DMLab.

**SimCore**   (Gregor et al., 2019) studied the memory abilities of agents by probing representations and compared a range of agent objectives and memory mechanisms, an approach that we build upon in this paper. However, their datasets and implementations were not released, making it difficult for the research community to build upon the work. A standardized probe benchmark is available for Atari (Anand et al., 2019), but those tasks require almost no memory.

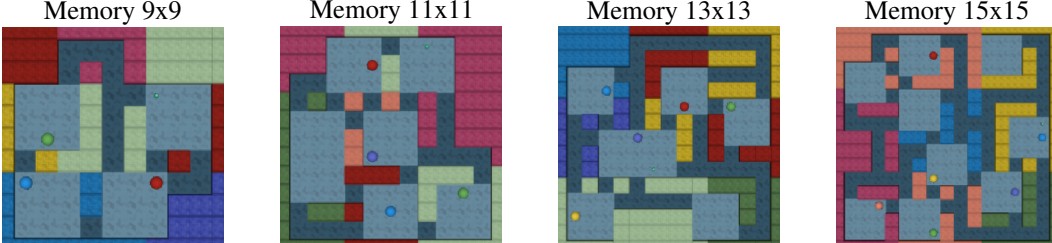

| Memory 9x9 | Memory 11x11 | Memory 13x13 | Memory 15x15 |

**Figure 2:** Examples of randomly generated Memory Maze layouts of the four sizes.

**DM Memory Suite** (Fortunato et al., 2019) consists of 5 existing DMLab tasks and 7 variations of T-Maze and Watermaze tasks implemented in the Unity game engine, which neccessitates interfacing with a provided Docker container via networking. These tasks pose an exploration challenge due to the initialization far away from the goal, creating a confounding factor in agent performance. Moreover, the tasks tend to require only small memory capacity, namely 1 bit for T-Mazes and 1 coordinate for Watermazes.

## 3 THE MEMORY MAZE

Memory Maze is a domain of randomized mazes specifically designed for evaluating the long-term memory abilities of RL agents. The agent navigates in the maze using first-person camera observations of the 3D scene while moving on a 2D planar layout. Memory Maze isolates long-term memory from confounding agent abilities, such as exploration, and requires remembering several pieces of information: the positions of objects, the wall layout, and the agent's own position. This section introduces three aspects of the benchmark: (1) an online reinforcement learning environment with four tasks, (2) an offline dataset, and (3) a protocol for evaluating representations on this dataset by probing.

### 3.1 ENVIRONMENT

The Memory Maze environment is implemented using MuJoCo (Todorov et al., 2012) as the physics and graphics engine and the dm_control (Tunyasuvunakool et al., 2020) library for building RL environments. The environment can be installed as a pip package `memory-maze` or from the source code, available on the project website [1]. There are four Memory Maze tasks with varying sizes and difficulty: Memory 9x9, Memory 11x11, Memory 13x13, and Memory 15x15.

The task is inspired by a game known as scavenger hunt or treasure hunt. The agent starts in a randomly generated maze containing several objects of different colors. The agent is prompted to find the target object of a specific color, indicated by the border color in the observation image. Once the agent finds and touches the correct object, it gets a +1 reward, and the next random object is chosen as a target. If the agent touches the object of the wrong color, there is no effect. Throughout the episode, the maze layout and the locations of the objects do not change. The episode continues for a fixed amount of time, so the total episode return is equal to the number of targets the agent can find in the given time. See Figure 1 for an illustration.

The episode return is inversely proportional to the average time it takes for the agent to locate the target objects. If the agent remembers the location of the prompted object and how the rooms are connected, the agent can take the shortest path to the object and thus reach it quickly. On the other hand, an agent without memory cannot remember the object position and wall layout and thus has to randomly explore the maze until it sees the requested object, resulting in several times longer duration. Thus, the score on the Memory Maze tasks correlates with the ability to remember the maze layout, particularly object locations and paths to them.

Memory Maze sidesteps the hard exploration problem present in many T-Maze and Watermaze tasks. Due to the random maze layout in each episode, the agent will sometimes spawn close to the object of the prompted color and easily collect the reward. This allows the agent to quickly bootstrap to a policy that navigates to the target object once it is visible, and from that point, it can improve by developing memory. This makes training much faster compared to, for example, the "Spot the

---

[1]https://github.com/jurgisp/memory-maze

| Parameter | Memory 9x9 | Memory 11x11 | Memory 13x13 | Memory 15x15 |
|---|---|---|---|---|
| Number of objects | 3 | 4 | 5 | 6 |
| Number of rooms | $3-4$ | $4-6$ | $5-6$ | 9 |
| Room size | $3-5$ | $3-5$ | $3-5$ | 3 |
| Episode length (steps at 4Hz) | 1000 | 2000 | 3000 | 4000 |
| Mean maximum score (oracle) | 34.8 | 58.0 | 74.5 | 87.7 |

**Table 1:** Memory Maze environment details.

Difference" set of tasks in DM Memory Suite that require the agent to cross a long corridor to receive any reward signal.

The sizes are designed such that the Memory 15x15 environment is challenging for a human player and out of reach for state-of-the-art RL algorithms, whereas Memory 9x9 is easy for a human player and solvable with RL, with 11x11 and 13x13 as intermediate stepping stones. See Table 1 for details and Figure 2 for an illustration.

## 3.2 OFFLINE DATASET

We collect a diverse offline dataset of recorded experience from the Memory Maze environments. This dataset is used in the present work for the offline probing benchmark and also enables other applications, such as offline RL.

We release two datasets: Memory Maze 9x9 (30M) and Memory Maze 15x15 (30M). Each dataset contains 30 thousand trajectories from Memory Maze 9x9 and 15x15 environments respectively. A single trajectory is 1000 steps long, even for the larger maze to increase the diversity of mazes included while keeping the download size small. The datasets are split into 29k trajectories for training and 1k for evaluation.

The data is generated by running a scripted policy on the corresponding environment. The policy uses an MPC planner (Richards, 2005) that performs breadth-first-search to navigate to randomly chosen points in the maze under action noise. This choice of policy was made to generate diverse trajectories that explore the maze effectively and that form loops in space, which can be important for learning long-term memory. We intentionally avoiding recording data with a trained agent to ensure a diverse data distribution (Yarats et al., 2022) and to avoid dataset bias that could favor some methods over others.

The trajectories include not only the information visible to the agent – first-person image observations, actions, rewards – but also additional semantic information about the environment, including the maze layout, agent position, and the object locations. The details of the data keys are in Table 2.

## 3.3 OFFLINE PROBING

Unsupervised representation learning aims to learn representations that can later be used for downstream tasks of interest. In the context of partially observable environments, we would like unsupervised representations to summarize the history of observations into a representation that contains information about the state of the environment beyond what is visible in the current observation by remembering salient information about the environment. Unsupervised representations are commonly evaluated by probing (Oord et al., 2018; Chen et al., 2020; Gregor et al., 2019; Anand et al., 2019), where a separate network is trained to predict relevant properties from the frozen representations.

We introduce the following four Memory Maze offline probing benchmarks: Memory 9x9 Walls, Memory 15x15 Walls, Memory 9x9 Objects, and Memory 15x15 Objects. These are based on either using the maze wall layout (`maze_layout`) or agent-centric object locations (`targets_vec`) as the probe prediction target, trained and evaluated on either Memory Maze 9x9 (30M) or Memory Maze 15x15 (30M) offline datasets.

The evaluation procedure is as follows. First, a sequence representation model (which may be a component of a model-based RL agent) is trained on the offline dataset with a semi-supervised loss based on the first-person image observations conditioned by actions. Then a separate probe network is trained to predict the probe observation (either maze wall layout or agent-centric object locations) from the internal state of the model. Crucially, the gradients from the probe network are not propagated into the model, so it only learns to decode the information already present in the

| Key | Shape | Type | Description |
|---|---|---|---|
| image | $(64, 64, 3)$ | uint8 | First-person view observation |
| action | $(6)$ | binary | Last action, one-hot encoded |
| reward | $()$ | float | Last reward |
| maze_layout | $(9, 9) \mid (15, 15)$ | binary | Maze layout (wall / no wall) |
| agent_pos | $(2)$ | float | Agent position in global coordinates |
| agent_dir | $(2)$ | float | Agent orientation as a unit vector |
| targets_pos | $(3, 2) \mid (6, 2)$ | float | Object locations in global coordinates |
| targets_vec | $(3, 2) \mid (6, 2)$ | float | Object locations in agent-centric coordinates |
| target_pos | $(2)$ | float | Current target object location, global |
| target_vec | $(2)$ | float | Current target object location, agent-centric |
| target_color | $(3)$ | float | Current target object color RGB |

**Table 2:** Entries in the offline dataset. The tensors are saved as NPZ files with an additional time step, e.g. image tensor for a 1000-step long trajectory is $(1001, 64, 64, 3)$. The first element is the image *before* the first action and the last element is the image *after* the last action.

internal state, but it does not drive the representation. Finally, the predictions of the probe network are evaluated on the hold-out dataset. When predicting the wall layout, the evaluation metric is prediction accuracy, averaged across all tiles of the maze layout. When predicting the object locations, the evaluation metric is the root-mean-squared error (RMSE), averaged over the objects. The final score is calculated by averaging the evaluation metric over the *second half* (500 steps) of each trajectory in the evaluation dataset. This is done to remove the initial exploratory part of each trajectory, during which the model has no way of knowing the full layout of the maze (see Figure C.1). We make this choice so that a model with perfect memory could reach $0.0$ RMSE on the Objects benchmark and $100\%$ accuracy on the Walls benchmark.

The architecture of the probe network is defined as part of the benchmark to ensure comparability: it is an MLP with 4 hidden layers, 1024 units each, with layer normalization and ELU activation after each layer (see Table E.3). The input to the probe network is the representation of the model — which should be a 1D vector of length 2048 — concatenated with the position and orientation of the agent. The agent coordinate is provided as additional input because the wall layout prediction target is presented in the global grid coordinate system, which the agent has no way of knowing from the first-person observations. The probe network is trained with BCE loss when the output is wall layout, and with MSE loss when the output is object coordinates.

We found a linear probe to not be suitable for our study because it is not powerful enough to extract the desired features (such as the wall layout) from the recurrent state. There is a potential concern that a powerful enough probe network could extract any desired information about the history of observations from the recurrent state, which would invalidate the logic of the test. We provide evidence in Appendix D that this is not the case, and the choice of the 4-layer network is justified.

## 4   ONLINE EXPERIMENTS

In this section, we present online RL benchmark results on the four Memory Maze tasks (9x9, 11x11, 13x13, 15x15) introduced in Section 3.1. We have evaluated the following baselines, including a strong model-based and model-free RL algorithm each:

- **Human Player**    The data was collected from one person who had experience playing first-person computer games but had not previously seen the Memory Maze task. The player first played several episodes to familiarize themselves with the task and GUI. They then played the recorded episodes, for which they were instructed to concentrate and strive for the maximum score. We recorded ten episodes for each maze size.

- **Dreamer**    For a model-based RL baseline we evaluated the DreamerV2 agent (Hafner et al., 2020). We mostly used the default parameters from the Atari agent in (Hafner et al., 2020) but increased the RSSM recurrent state size and tuned the KL scale and entropy scale. The same parameters were used across all four tasks. We increased the speed of data generation (environment steps per update) to make the training faster in wall clock time, at some expense to the sample

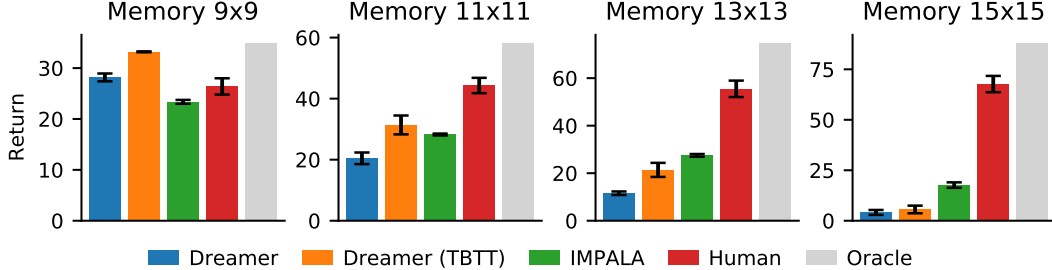

**Figure 3:** Online RL benchmark results after 100M environment steps of training. Error bars show the standard deviation over five runs. We find that current algorithms benefit from training with truncated backpropagation through time and succeed on small mazes, but fall short of human performance on the large mazes, leaving room for future algorithmic designs to be evaluated on the Memory Maze.

| Method | Memory 9x9 | Memory 11x11 | Memory 13x13 | Memory 15x15 |
|--------|-----------|--------------|--------------|--------------|
| Oracle | 34.8 | 58.0 | 74.5 | 87.7 |
| Human | $26.4_{\pm 1.6}$ | $44.3_{\pm 2.5}$ | $55.5_{\pm 3.5}$ | $67.7_{\pm 4.0}$ |
| IMPALA | $23.4_{\pm 0.4}$ | $28.2_{\pm 0.3}$ | $\mathbf{27.5}_{\pm 0.6}$ | $\mathbf{17.7}_{\pm 1.4}$ |
| Dreamer | $28.2_{\pm 0.8}$ | $20.4_{\pm 1.9}$ | $11.6_{\pm 0.7}$ | $4.1_{\pm 1.2}$ |
| Dreamer (TBTT) | $\mathbf{33.2}_{\pm 0.1}$ | $\mathbf{31.4}_{\pm 3.1}$ | $21.4_{\pm 3.0}$ | $5.6_{\pm 1.9}$ |

**Table 3:** Online RL benchmark results. The RL agents were trained for 100M steps, and the reported scores are averaged over five runs, with the standard deviation indicated as a subscript. Human score is an average of 10 episodes, with the bootstrapped standard error of the mean shown as a subscript.

efficiency. We trained Dreamer on 8 parallel environments. The full hyperparameters are listed in Table E.1.

- **Dreamer (TBTT)** Truncated backpropagation through time (TBTT) is known to be an effective method for training RNNs to preserve information over time. For example, R2D2 agent (Kapturowski et al., 2018), when trained with the stored state, shows significant improvement on DMLab tasks compared to zero state initialization. Original Dreamer is trained with zero input state on each batch, which may limit its ability to form long-term memories. To test this, we implement TBTT training in Dreamer by replaying complete trajectories sequentially and passing the RSSM state from one batch to the next. We alleviate the potential problem of correlated batches by forming each $T \times B$ batch from $B$ different episodes, and we start replaying the first episode from a random offset to avoid synchronized episode starts.

- **IMPALA** We choose IMPALA (V-trace) as a strong model-free baseline, which performs well on DMLab-30 (Espeholt et al., 2018) and has a reliable implementation available in SEED RL (Espeholt et al., 2019). The hyperparameters used in our experiments are shown in Table E.2. We tuned the entropy loss scale, learning rate and Adam epsilon. We also tried increasing the LSTM size, but it did not improve performance. The efficient implementation allowed us to use 128 parallel environments, which was important for the performance achieved on the larger mazes.

- **Oracle** We establish an upper bound on the task scores by training an oracle agent that observes complete information about the maze layout and follows the shortest path to the target object. Note that no real agent relying on first-person observations can achieve this upper bound score because the oracle receives the maze layout as input from the beginning of the episode without having to explore it over time. However, we can estimate that this initial exploration, if done efficiently, should not take more steps than reaching a few targets. So the maximum achievable score for agents is within a few points of the Oracle upper bound.

In the future, it would be interesting to evaluate additional baselines on Memory Maze, such as MRA (Fortunato et al., 2019) that uses episodic memory and GTrXL (Parisotto et al., 2020) that uses a transformer architecture. Unfortunately, the source code of these agents is not publicly available, so we were unable to include them in our comparison.

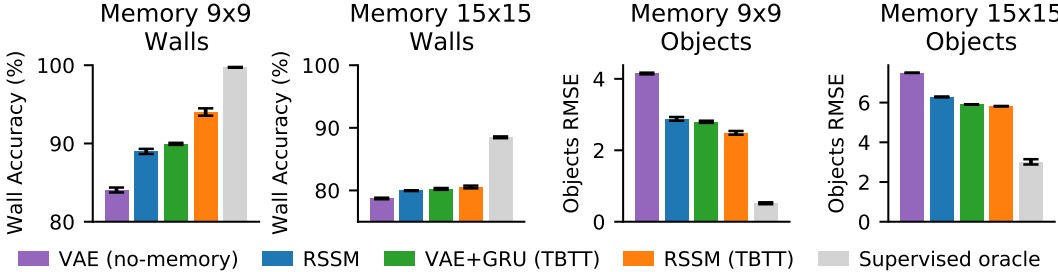

**Figure 4:** Offline probing results on Memory 9x9 and Memory 15x15 datasets. Left: Average accuracy of wall probing (higher is better), with the perfect score being $100\%$ and VAE indicating a no-memory baseline. Right: Average root-mean-squared error (RMSE) of object probing (lower is better), measured in grid cell units, with the perfect score being 0 and VAE indicating a no-memory baseline.

| Method | Memory 9x9 Walls | Memory 15x15 Walls | Memory 9x9 Objects | Memory 15x15 Objects |
|---|---|---|---|---|
| Constant baseline | 80.8 | 78.3 | 4.9 | 8.0 |
| VAE (no memory) | $84.1_{\pm 0.3}$ | $78.7_{\pm 0.1}$ | $4.1_{\pm 0.1}$ | $7.5_{\pm 0.1}$ |
| Supervised oracle | $99.7_{\pm 0.1}$ | $88.5_{\pm 0.1}$ | $0.5_{\pm 0.1}$ | $3.0_{\pm 0.1}$ |
| RSSM | $89.0_{\pm 0.3}$ | $80.0_{\pm 0.1}$ | $2.9_{\pm 0.1}$ | $6.3_{\pm 0.1}$ |
| VAE+GRU (TBTT) | $90.0_{\pm 0.1}$ | $80.2_{\pm 0.1}$ | $2.8_{\pm 0.1}$ | $5.9_{\pm 0.1}$ |
| RSSM (TBTT) | $\mathbf{94.0}_{\pm 0.5}$ | $\mathbf{80.5}_{\pm 0.2}$ | $\mathbf{2.5}_{\pm 0.1}$ | $\mathbf{5.8}_{\pm 0.1}$ |

**Table 4:** Offline probing benchmark results. The score is measured in wall prediction accuracy (%) for the Walls benchmarks and in RMSE (measured in grid cell units) for the Object benchmarks. The scores are calculated over the second half of the evaluation episodes to remove the effect of the initial memory burn-in (see Figure C.1). The models were trained for 1M gradient steps, and we report the mean score over three training runs, with the standard deviation indicated as a subscript.

All agents were evaluated after 100 million environment steps of training. For each baseline and task, we trained five agents with different random seeds and report the average scores. A single Dreamer training run took 14 days to train using one GPU learner and 8 CPU actors. A single IMPALA training run took 20 hours to train using one GPU learner and 128 CPU actors. Our experimental results are summarized in Figure 3 and Table 3. The training curves are provided in Appendix A.

First, we observe that the task is challenging but solvable for a human player. The mean human score is approximately 75% of the oracle upper bound across all four maze sizes. Inspection of episode replays shows that the player is slow at collecting the first few rewards while exploring the new unknown layout, after which the remaining rewards are collected relatively quickly (see Appendix B). This indicates that the task indeed relies on the ability to remember. Moreover, the learning phase becomes longer in the larger mazes.

Second, we note that the performance of RL agents exceeds the human performance on the smallest 9x9 maze but is far below the human baseline on the largest 15x15 maze, with 11x11 and 13x13 interpolating between the two extremes. This shows that our benchmark uncovers the limits of the current RL methods and enables measurable progress for future algorithms.

Third, we observe the utility of training Dreamer with TBTT, which shows a clear boost to the original Dreamer across all maze sizes. For example, on the Memory Maze 9x9, only the Dreamer (TBTT) agent achieves near-optimal performance.

Finally, even though Dreamer (TBTT) outperforms model-free IMPALA on the smaller mazes, on the larger mazes IMPALA is the best RL agent, making steady progress even on the Memory Maze 15x15. We speculate that IMPALA is better at remembering task-relevant information for longer because model-free policy training encourages the encoder and RNN to only process and retain task-relevant information and ignore the rest. In contrast, the world model of Dreamer tries to remember as much information about the environment as possible, which may limit the memory horizon.

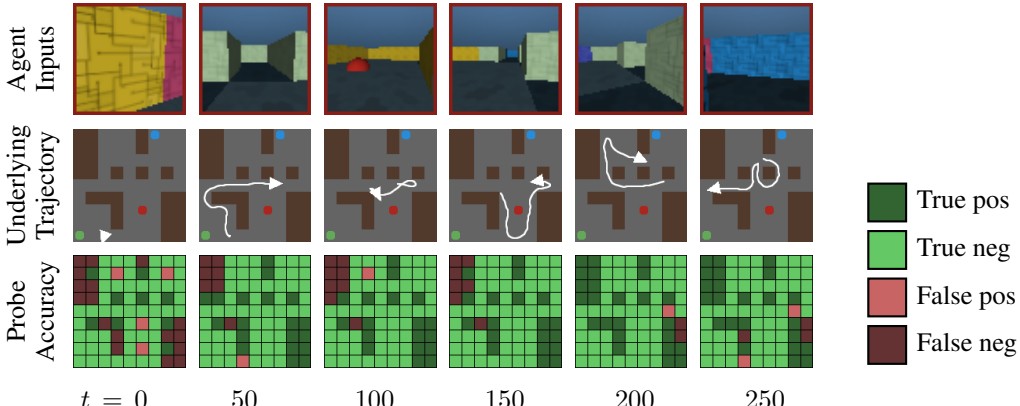

**Figure 5:** Probing the representations of a trained RSSM (TBTT) model for the information about the global layout of the maze. The model only sees raw pixels of the 3D first-person view (top row). The probe predictions of the wall layout are shown in the bottom row. After initial exploration, almost the full layout can be decoded from the representation in the small maze, demonstrating its ability to retain information over time.

## 5 OFFLINE EXPERIMENTS

In this section, we present offline probing experiments on the four benchmarks: Memory 9x9 (Walls), Memory 9x9 (Objects), Memory 15x15 (Walls), and Memory 15x15 (Objects), that were introduced in Section 3.3. We trained and evaluated the following sequence representation learning models:

- **RSSM**  Recurrent State Space Model (Hafner et al., 2018) is the world model component of Dreamer agent. We use the exact same model that is part of the DreamerV2 agent (Hafner et al., 2020) that was evaluated on the online benchmark, with the hyperparameters listed in Table E.1. The probe network receives the full internal state as input, which is a concatenation of deterministic and stochastic states of RSSM.

- **RSSM (TBTT)**  As in the online experiments, we evaluate the effect of training RSSM with truncated backpropagation through time (TBTT). Instead of starting with zero recurrent state on each training batch, we sample training sequences from trajectories sequentially and carry over the state from one batch to the next. Note that during inference time, both RSSM and RSSM (TBTT) propagate the state over the complete 1000-step trajectory. The difference is only during training, where RSSM effectively resets the state to zero every 48 steps, and RSSM (TBTT) does not.

- **VAE+GRU (TBTT)**  This baseline is a recurrent model that operates on separately learned representations (Ha and Schmidhuber, 2018). The image embeddings are trained with a standard VAE (Kingma and Welling, 2013) on the full dataset, without consideration of their sequential ordering. A GRU-based RNN summarizes the sequence of embeddings and actions in a recurrent state, and a dynamics MLP predicts the next observation embedding, given the current state and action. This network is trained with an MSE loss for the next-embedding prediction. The hyperparameters are chosen to match RSSM where relevant (see Table E.4). The probe predictor uses the GRU hidden state as the representation. This model is also trained with TBTT.

- **Supervised Oracle**  For comparison, we evaluate a baseline trained in a supervised manner, allowing the gradients of the probe prediction loss to propagate into the whole model. We use the exact same architecture as VAE+GRU, with the only difference of not stopping the probe gradients during training. It is an "oracle" in the sense that it does not follow the standard evaluation procedure, and uses hidden information (probe observations) to train the representation model. This baseline is useful as an upper (optimistic) bound for models with similar architecture.

- **VAE (no-memory)**  A model that simply uses the VAE image embeddings as the input to the probe prediction. It has no memory by design, and it is able to only predict the part of the probe observation which can be inferred from the current first-person view. It serves as a lower (pessimistic) bound for memory models.

All models were trained for 1M gradient steps, and we repeated each training with three random seeds. The evaluation results are summarized in Figure 4 and Table 4. In addition to the trained models, we include a Constant baseline that outputs the training set mean prediction. This score is relatively

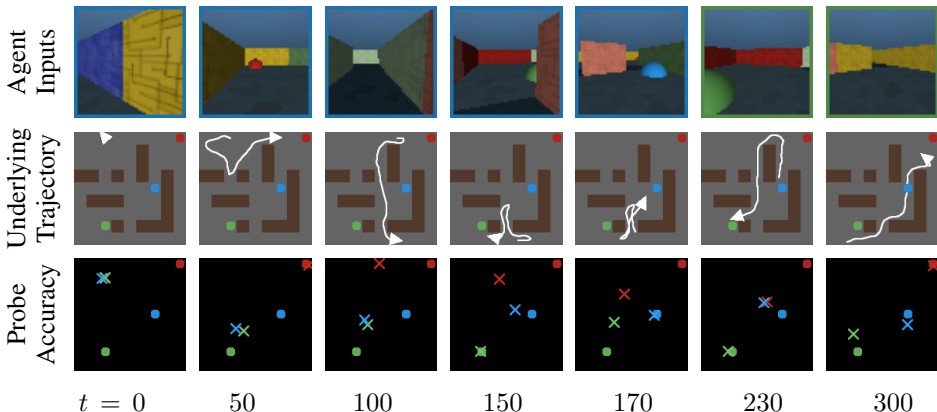

**Figure 6:** Probing the representations of a trained RSSM (TBTT) model for the object locations. The model only sees raw pixels of a 3D first-person view (top row). The bottom row shows the object locations as predicted from the frozen representation (x) versus the actual locations (o). The output of the probe decoder is the agent-centric object locations, here they are transformed into the global coordinates for visualization.

high on Walls (e.g. $80.8\%$ for 9x9) because a relatively small number of grid cells vary between different layouts. The performance of models falls on the scale between the Constant baseline, as the minimum, and $100\%$ accuracy or $0.0$ RMSE as the maximum.

We observe that all evaluated models show some memory capacity (being better than the no-memory baseline), but fall short of perfect memory. The 15x15 Walls is an especially challenging benchmark, where the models barely outperform the no-memory baseline. This is consistent with the low performance of the corresponding agents on the 15x15 online RL benchmark, and suggests that Memory Maze can be a fruitful platform for further research, both in the online and offline settings.

Among the models, RSSM (TBTT) reaches the highest performance, outperforming RSSM and VAE+GRU (TBTT). This shows that both truncated backpropagation through time and the stochastic bottleneck are helpful ingredients for learning to remember. On Memory 9x9 (Walls), RSSM (TBTT) correctly predicts $94\%$ of the wall layout after the initial burn-in phase, compared to the $84.1\%$ no-memory baseline. This is consistent with the fact that Dreamer (TBTT) achieves near-perfect performance on the Memory 9x9 online RL benchmark. An illustration of a sample evaluation trajectory, observations, and the corresponding probe predictions is presented in Figure 5 and Figure 6.

## 6 DISCUSSION

We introduced Memory Maze, an environment for evaluating the memory abilities of RL agents. We conduced a human study to confirm that the tasks are challenging but solvable for a human player and found that human rewards increase within each episode, demonstrating the need to build up memory to solve the tasks. We collected a diverse offline dataset that includes semantic information to evaluate learned representations through probing, by predicting both task-specific and task-agnostic information. Empirically, we find that truncated backpropagation through time offers a significant boost in probe performance, further confirming that memory is the key challenge in this benchmark. An evaluation of a strong model-free and model-based RL algorithm each show learning progress on smaller mazes but fall short of human performance on larger mazes, presenting a challenge for future algorithms research.

**Acknowledgements** We thank Google Research infrastructure team to provide JP with access to computational resources. DH was at Google Research for part of the project before starting an internship at DeepMind.

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

# A    ONLINE TRAINING CURVES

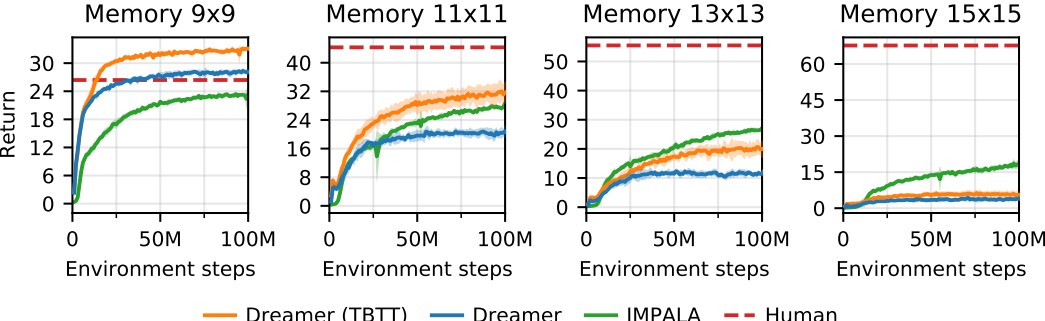

**Figure A.1:** Training curves on the Memory Maze tasks. The scores are averaged over five runs, standard deviation is shown as a shaded area. After training for 100M environment steps, Dreamer (TBTT) shows the best performance on the two smaller benchmarks Memory 9x9 and 11x11, and IMPALA is the best on the larger mazes 13x13 and 15x15.

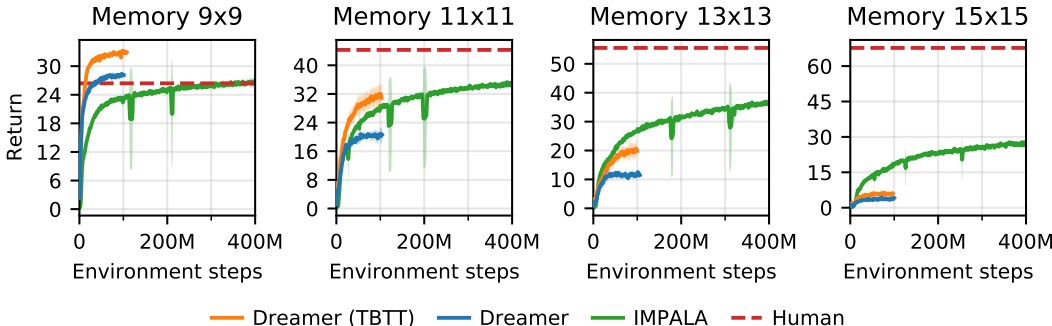

**Figure A.2:** Extended IMPALA training curves on Memory Mazes over 400M steps. On Memory Maze 15x15 it makes steady progress beyond 100M steps reaching score of $27.7 \pm 0.2$, but it is flattening out far below the Human baseline.

# B    HUMAN SCORES

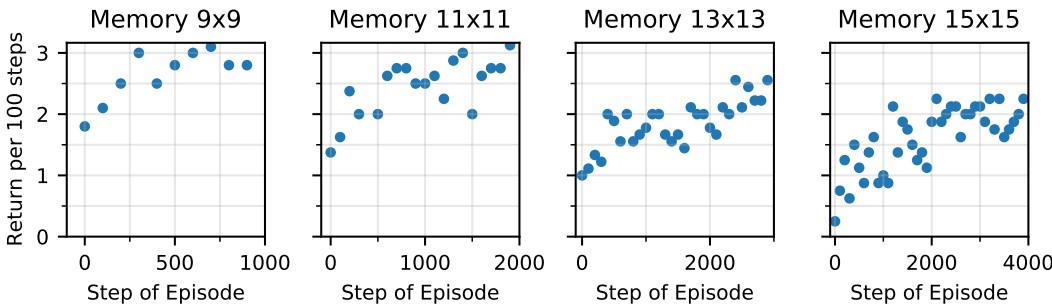

**Figure B.1:** The average reward collected as a function of the episode step within the episode, summed over 100 step bins, averaged over the episodes of the human player. The reward collection is slow in the beginning of the episode, when the maze layout is unknown and the player has to explore the maze. The pace increases as the player memorizes the maze and object locations, and can effectly navigate between them. This shows that the task relies on long-term memory.

# C   EXTRA OFFLINE PROBING RESULTS

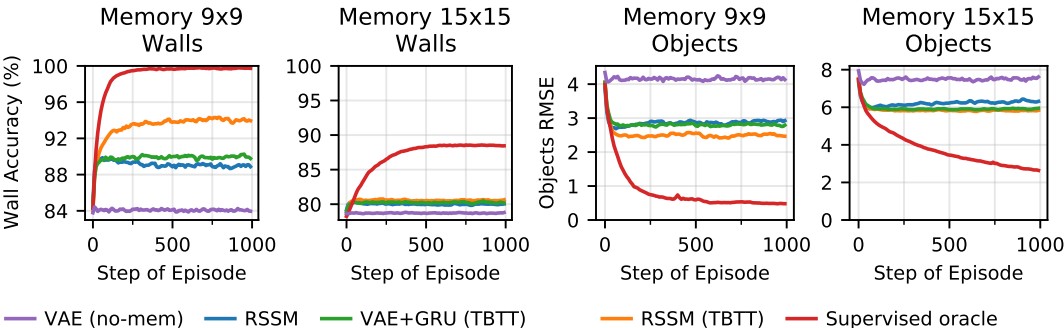

**Figure C.1:** Offline probe prediction performance as a function of the episode step. The prediction error starts high at the beginning of the episode, when the layout of the maze is impossible to predict, and then drops as the agent explores the maze. The benchmark results reported in Table 4 are equal to the mean value of these plots over the range from 500 to 1000, where the prediction error is stable.

# D   PROBE NETWORK

Here we motivate the choice of using a 4-layer MLP probe network as opposed to a more typical linear probe. Figure D.1 shows how probe accuracy on the Memory Maze 9x9 (Walls) benchmark depends on the number of layers in the MLP. The linear probe performs poorly; the accuracy increases up to 4 layers. More layers provide little additional gain. Note the probe has to take into account the input about the position of the agent, which is infeasible with a linear probe. At the other extreme, one may worry that a powerful enough network could extract any desired information about the history of observations, but we show that 1) additional layers don't do better and 2) a probe MLP trained on a random untrained RSSM is only marginally better than the constant baseline (80.8%):

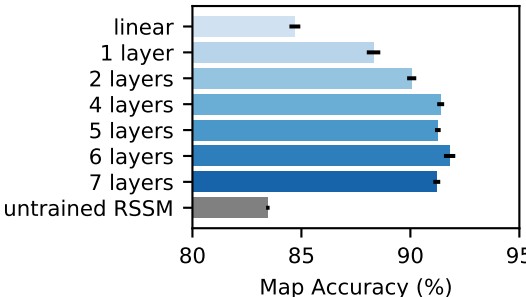

**Figure D.1:** Average map probe accuracy on Memory Maze 9x9 (Walls) offline benchmark as a function of the number of layers in the probe MLP. The last row shows the performance of a 4-layer MLP probe trained to decode information from a randomly initialized RSSM.

# E    HYPERPARAMETERS

For tuning hyperparameters we performed one-dimensional grid searches for one parameter at a time. We evaluated parameters on the Memory 11x11 environment, since it is the smallest challenging one, and then used the best values across all environments. For Dreamer agent we scanned over the following parameter values: recurrent state size (512, 1024, 2048, 4096), KL scale (0.1, 0.3, 1.0, 3.0), entropy scale ($3 \cdot 10^{-4}$, $1 \cdot 10^{-3}$, $3 \cdot 10^{-3}$, $1 \cdot 10^{-2}$). For IMPALA we tuned the entropy scale ($3 \cdot 10^{-4}$, $1 \cdot 10^{-3}$, $3 \cdot 10^{-3}$, $1 \cdot 10^{-2}$), learning rate ($1 \cdot 10^{-4}$, $2 \cdot 10^{-4}$, $3 \cdot 10^{-4}$, $4 \cdot 10^{-4}$) and Adam epsilon ($3 \cdot 10^{-9}$, $10^{-7}$, $10^{-6}$, $10^{-5}$).

| Parameter | Value |
|---|---|
| **Model** | |
| RSSM recurrent state size | **2048** |
| RSSM stochastic discrete latent size | $32 \times 32$ |
| RSSM number of hidden units | **1000** |
| Decoder MLP number of layers | 4 |
| Decoder MLP number of units | 400 |
| Probe MLP number of layers | 4 |
| Probe MLP number of units | 1024 |
| **Objective** | |
| Discount | 0.995 |
| $\lambda$-target parameter | 0.95 |
| Actor entropy loss scale | 0.001 |
| RSSM KL loss scale | **1.0** |
| RSSM KL balancing | 0.8 |
| **Training** | |
| Parallel environments | 8 |
| Replay buffer size | **10M** steps |
| Batch size | **32** sequences |
| Sequence length | 48 |
| Imagination horizon | 15 |
| Environment steps per update | **25** |
| Slow critic update interval | 100 |
| World model learning rate | $\mathbf{3 \cdot 10^{-4}}$ |
| Actor learning rate | $\mathbf{1 \cdot 10^{-4}}$ |
| Critic learning rate | $1 \cdot 10^{-4}$ |
| AdamW epsilon | $10^{-5}$ |
| AdamW weight decay | $\mathbf{10^{-2}}$ |
| Gradient clipping | **200** |

**Table E.1:** Hyperparameters used when training Dreamer and Dreamer (TBTT) agents. The differences from the parameters of the original DreamerV2 Atari model (Table D.1 in Hafner et al. (2020)) are shown in bold face. The most substantial differences are: RSSM recurrent state size ($600 \rightarrow 2048$), replay buffer size ($10^6 \rightarrow 10^7$), environment steps per update ($4 \rightarrow 25$) and KL loss scale ($0.1 \rightarrow 1.0$).

| Parameter | Value |
|---|---|
| Parallel environments | 128 |
| LSTM size | 256 |
| Discount | 0.99 |
| $\lambda$-target parameter | 0.95 |
| Actor entropy loss scale | 0.001 |
| Batch size | 32 sequences |
| Sequence length | 100 |
| Adam learning rate | $2 \cdot 10^{-4}$ |
| Adam epsilon | $10^{-7}$ |

**Table E.2:** Hyperparameters used when training IMPALA agent. The parameters were based on those used in (Espeholt et al., 2019) on the DMLab-30 tasks, with the additional tuning of entropy scale, learning rate and Adam epsilon. We also tested larger LSTM size, but it provided no extra performance.

| Parameter | Value |
|---|---|
| Probe MLP number of layers | 4 |
| Probe MLP number of units | 1024 |
| Probe MLP activation | ELU |
| Probe MLP layer norm epsilon | $10^{-3}$ |
| Adam learning rate | $3 \cdot 10^{-4}$ |
| Adam epsilon | $10^{-5}$ |

**Table E.3:** Hyperparameters of the probe decoder used in the offline probing benchmarks.

| Parameter | Value |
|---|---|
| GRU state size | 2048 |
| VAE stochastic latent size | 32 |
| Encoder | (same as Dreamer) |
| Decoder | (same as Dreamer) |
| Dynamics MLP number of layers | 2 |
| Dynamics MLP number of units | 400 |
| VAE KL loss scale | 0.1 |
| Batch size | 32 sequences |
| Sequence length | 48 |
| AdamW learning rate | $3 \cdot 10^{-4}$ |
| AdamW epsilon | $10^{-5}$ |
| AdamW weight decay | $10^{-2}$ |
| Gradient clipping | 200 |

**Table E.4:** Hyperparameters used for the GRU+VAE offline probing baseline. Parameters are chosen to match Dreamer model Table E.1 where relevant.

# F    ENVIRONMENT DETAILS

| Index | Action | Forward/Backward | Left/Right |
|:---:|:---|:---:|:---:|
| 0 | noop | 0.0 | 0.0 |
| 1 | forward | +1.0 | 0.0 |
| 2 | turn left | 0.0 | −1.0 |
| 3 | turn right | 0.0 | +1.0 |
| 4 | forward left | +1.0 | −1.0 |
| 5 | forward right | +1.0 | +1.0 |

**Table F.1:** Memory Maze 6-dimensional discrete action space mapping to the underlying DMC continuous action space.

