# OpenReview forum: "Evaluating Long-Term Memory in 3D Mazes"
_ICLR.cc/2023/Conference — ICLR 2023 poster_

### Official Review · Reviewer_pb7h · 2022-10-24

**Confidence:** 4
**Correctness:** 4
**Technical Novelty And Significance:** 3
**Empirical Novelty And Significance:** 3
**Recommendation:** 8

**Clarity, Quality, Novelty And Reproducibility:**

**Clarity**

The presentation is uniformly clear. I found only a couple of problems:

1. “The hyperparameters are chosen to match RSSM were relevant”.  (“were” -> “where”?)

2. In Table F.1, index 3 probably refers to “turn right” instead of “turn left”.


**Quality**

The environment, benchmarks, and evaluations are of consistently high quality.

**Novelty**

There is little novelty in this work. Its value lies in satisfying a number of tricky requirements for well-focused tests of memory capabilities.

**Reproducibility**

The environments will be very accessible once available as a pip package and open source.


**Strength And Weaknesses:**

**Strengths**

Despite the ubiquity of partially observable environments, and many existing RL tasks that require some degree of memory, most fall short in ways detailed by this paper. The design of the Memory Maze environment is well-motivated, and carefully positioned with regards to prior work. The result is a compelling RL environment for evaluating the memory abilities of agents. The episodes are appropriately lengthy, to measure memories that persist for thousands of timesteps.

The range of maze sizes, rooms, and objects is crafted to enable graduated evaluation of agent abilities. The baseline agents are well chosen, and include human results. The Dreamer baseline results are particularly interesting, outperforming a standard model-free agent (Impala) on smaller mazes, and under-performing on larger mazes. And the human results in Fib B.1 show how much room there is for RL methods to improve.

**Weaknesses**

The paper would be stronger if it included (along with the baselines) a novel agent architecture that established state-of-the-art results on Memory Maze. Without that, the benchmark’s impact will depend on the extent to which the community chooses to use it.


**Summary Of The Paper:**

This work presents a new RL environment, Memory Maze, designed to evaluate an agent’s ability to maintain moderately long-term memories of observations, independent of the common challenges of exploration and credit assignment. A dataset of expert trajectories is provided to support offline RL research. Baseline (including human) results demonstrate that the benchmark spans the range from currently solvable to very challenging. Further support is provided for probing the agent’s representations.

**Summary Of The Review:**

While this work is not flashy, and does not establish new SOTA results, the benchmark and baselines are likely to prove valuable in research to improve the memory of RL agents, which currently falls dramatically short of human-level.

Update (Nov 18): I have read all reviews and author responses. I still view this work as a solid contribution. My score remains unchanged.

---

> ### Author Response · Authors · 2022-11-15
> **Response to Reviewer pb7h**
>
> Thank you for your comments and for highlighting the high quality of the presented environments, benchmarks, and evaluations.
>
> > The paper would be stronger if it included (along with the baselines) a novel agent architecture that established state-of-the-art results on Memory Maze. Without that, the benchmark’s impact will depend on the extent to which the community chooses to use it.
>
> We agree that this is a fruitful direction, and we plan to pursue it in future research. Regardless, thank you for appreciating the value of publishing the benchmark, which allows the community to work on novel agents in parallel.
>
> > “The hyperparameters are chosen to match RSSM were relevant”. (“were” -> “where”?) / In Table F.1, index 3 probably refers to “turn right” instead of “turn left”.
>
> Thank you for spotting these, we have fixed them in the paper.

---

> > ### Comment · Reviewer_pb7h · 2022-11-17
> > **Reply**
> >
> > Thank you!

---

### Official Review · Reviewer_9HKy · 2022-10-28

**Confidence:** 4
**Correctness:** 3
**Technical Novelty And Significance:** 2
**Empirical Novelty And Significance:** 4
**Recommendation:** 8

**Clarity, Quality, Novelty And Reproducibility:**

The clarity is quite good overall.

Conceptual originality is perhaps not that huge since many variants of the basic theme "find objects in a maze" have been proposed to test memory and/or exploration (for example) before. However, I believe carefully thought out and evaluated environments are very important for progress, so I don't think this is a crucial issue here. To the best of my knowledge, the empirical evaluation conducted here is original and I feel that that is enough.

Reproducibility is good (assuming the environment will actually be open-sourced as stated since I believe the evaluated agents are also open source and the hyperparameters are available in the appendix.

**Strength And Weaknesses:**

Strengths
=========
* Good benchmarks to test specific agent capabilities facilitate progress in reinforcement learning research.
* The proposed environment seems well thought out and it's generally well explained how it aims to test memory specifically while mitigating dependency on other challenges like exploration.
* The paper is for the most part very well-written and clear.
* The empirical results are interesting and seem generally well done. They do a good job of highlighting the utility of the environment for evaluating the memory capabilities of strong reinforcement learning agents. I like the inclusion of the offline probing task and associated empirical results as a way to evaluate memory capability separated from the other challenges of reinforcement learning within the same environment.

Weaknesses
==========
* Certain claims are not clearly demonstrated in the paper or are overly vague. For example "This makes training much faster compared to, for example, DM Memory Suite", while perhaps plausible, this is not meaningfully demonstrated. Moreover, this statement refers to a whole suite of tasks and I don't believe it's true for all of them so this should at least be clarified and ideally demonstrated in some manner. Another example is the statement "on the larger mazes IMPALA is the best RL agent" and the associated speculative explanation. This doesn't seem like a fair comparison at all since dreamer is trained with 8 actors and IMPALA with 128. Does this not mean IMPALA gets 16 times more data than dreamer? Of course, there is a lot of nuance to this since dreamer is also much slower to train, however, I don't think the way this is currently explained is particularly insightful.
* Human data is collected using only one person. Evaluating a larger group of people could allow measurement of within-group variance and gives a better sense of the consistency of human performance on this task. This also makes the use of the plural in "We conducted a human study to confirm that the tasks are challenging but solvable for average human players" debatably incorrect.
* Some empirical details such as how precisely hyperparameter tuning was done are missing. For example, for Dreamer, it just says they "tuned the KL and entropy scales to the new environment", tuned how (grid search, manual tweaking...)? On which version of the environment (since this hyperparameter is shared across maze sizes)? For Impala "we tuned the entropy loss scale by scanning different values and then used the same hyperparameters across all four environments", scanning what values? And again for which version of the environment?
* Technical novelty is perhaps not that high, since previous work (for example the work of Wayne et. al. (2018), which is not cited) has investigated training agents on very similar tasks where an agent must repeatedly find objects in a randomized maze. However, given the focus of this work is on demonstrating the utility of the environment itself, which I think is worthwhile, I don't see this as a huge issue. On that note, it would be good to add a citation to Wayne et. al. (2018) as they also propose several tasks for evaluating memory, including one in which episodes consist of repeatedly locating a specific item in a randomized maze (albeit I believe only one item and not multiple as in Memory Maze).

Minor Comments and Corrections
==============================
* In Introduction (page 1): "solve later solve"
* In Introduction (page 2): "this offers and estimate"
* Section 4 (page 6): "...indicating that the human player had to observe the object positions multiple times before remembering them.", I don't quite understand how
* I can't see that code is made available in the present draft, though it is stated that "we open source the environment and make it easy to install and use". I think acceptance should definitely be conditional on this release given the testbed itself is the central contribution.
* Section 5 (page 8): "RSSM were relevant"
* For the MSE shown in Figure 4, it would be good to give some reference to understand the magnitude of these values relative to the maze (is each maze cell one unit?). Additionally, it would be good to show root MSE, or simply mean distance, instead of MSE so that one can more easily relate these errors to distances in the maze.

Reference
=========
Wayne, Greg, et al. "Unsupervised predictive memory in a goal-directed agent." arXiv preprint arXiv:1803.10760 (2018).

**Summary Of The Paper:**

This paper introduces a new benchmark, Memory Maze, for evaluating the long-term memory capabilities of RL agents. MemoryMaze consists of randomly generated 3D mazes wherein an agent gets only local camera observations along with a specified goal item to locate. In each episode, the environment presents the agent with a sequence of random target items in a random maze, and the agent receives a reward for each target successfully reached before a time limit runs out. The task emphasizes memory because after observing an item's location the first time, the agent should be able to remember it, along with the maze layout, to find it more quickly the next time within the same episode. Various maze sizes are tested to tune the difficulty of the memory challenge. The orthogonal exploration challenge is mitigated by the fact that the start location is randomized in each episode, thus the agent will occasionally be able to spawn close to a goal item to observe reward. Experiments benchmark the performance of a human against strong reinforcement learning agents. The first baseline is model-free IMPALA, the second is model-based Dreamerv2. For DreamerV2, one variant is tested where the RNN state is zeroed at the start of each training sequence, and another is tested where the RNN state is maintained for whole episodes, but gradients are propagated only for a fixed sequence length (called truncated backpropagation through time in the paper). The result indicated that, in the smaller mazes, strong RL agents can perform similarly to or better than humans, while for larger mazes the RL agents fall far behind, indicating room for future progress. In addition to the RL experiments, offline probing experiments are included which run sequence models on prerecorded episodes and probe the latent state of a sequence model for information about the location of items and walls in the maze.

**Summary Of The Review:**

This paper presents a well-thought-out benchmark for evaluating the long-term memory ability of current reinforcement learning systems. I believe this is an important contribution, as such environments are necessary for progress. Experiments are generally good and give insight into the utility of the environment for the stated purpose. My main concerns are a few issues with unclear claims and empirical details that could be added. The technical novelty is also not high as similar environments have been used to test memory before, but I still see this paper's aim of proposing and evaluating the environment itself as a worthwhile contribution.

---

> ### Author Response · Authors · 2022-11-15
> **Response to Reviewer 9HKy (part 1/2)**
>
> Thank you for your detailed feedback!
>
> > Certain claims are not clearly demonstrated in the paper or are overly vague. For example "This makes training much faster compared to, for example, DM Memory Suite", while perhaps plausible, this is not meaningfully demonstrated.
>
> We agree that this claim was too generic. We were referring, in particular, to the “Spot the Difference” set of tasks within the DM Memory Suite, that require the agent to cross a long corridor to receive any reward signal. We have adjusted the claim in the paper.
>
> > [...] another example is the statement "on the larger mazes IMPALA is the best RL agent" and the associated speculative explanation. This doesn't seem like a fair comparison at all since Dreamer is trained with 8 actors and IMPALA with 128. Does this not mean IMPALA gets 16 times more data than dreamer?
>
> Both Dreamer and IMPALA were trained for an equal amount of 100M environment steps (see Figure A.1), so in that respect comparison is fair. The difference lies only in the number of actors (parallel environment instances) hyperparameter, which is motivated by the different nature of the two algorithms:
>
> - IMPALA is an on-policy algorithm and thus performs only 1 gradient step on each collected trajectory. Therefore, its wall-clock time is more heavily influenced by data collection and benefits from a large number of actors. We used 128 actors, which is the default in the SEED-RL implementation.
> - DreamerV2 relies on off-policy training and thus spends more wall-clock time on updating the model on the replay buffer. The official implementation used only 1 actor, which we increased to 8 to speed up data collection to the point where the overall runtime was dominated by gradient steps.
>
> We chose the two algorithms because of their different computational and algorithm characteristics and because they are popular instances of on-policy and off-policy algorithms respectively, to provide representative baselines for our benchmark.
>
> > Human data is collected using only one person. Evaluating a larger group of people could allow measurement of within-group variance and gives a better sense of the consistency of human performance on this task. This also makes the use of the plural in "We conducted a human study to confirm that the tasks are challenging but solvable for average human players" debatably incorrect.
>
> Thank you for pointing this out. We have corrected the formulation, to not imply multiple subjects. We agree that a larger sample of diverse human players would be ideal and will consider including data from additional players in the final version of the paper.
>
> > Some empirical details such as how precisely hyperparameter tuning was done are missing. For example, for Dreamer, it just says they "tuned the KL and entropy scales to the new environment", tuned how (grid search, manual tweaking...)? On which version of the environment (since this hyperparameter is shared across maze sizes)? For Impala "we tuned the entropy loss scale by scanning different values and then used the same hyperparameters across all four environments", scanning what values? And again for which version of the environment?
>
> For tuning hyperparameters we performed grid searches for one parameter at a time, using Memory 11x11 for evaluation. We have updated the paper to include the details about values scanned in Appendix E.
>
> > [...] it would be good to add a citation to Wayne et. al. (2018) as they also propose several tasks for evaluating memory, including one in which episodes consist of repeatedly locating a specific item in a randomized maze (albeit I believe only one item and not multiple as in Memory Maze).
>
> Agreed! In fact, Wayne et. al. (2018) was an inspiration for this work. The citation got lost during editing, thanks for pointing it out.

---

> > ### Author Response · Authors · 2022-11-15
> > **Response to Reviewer 9HKy (part 2/2)**
> >
> >
> > > Section 4 (page 6): "...indicating that the human player had to observe the object positions multiple times before remembering them.", I don't quite understand how
> >
> > We agree that the statement was confusing and have replaced it with "...indicating that it takes humans longer to remember all object positions in the larger maze."
> >
> > > I can't see that code is made available in the present draft, though it is stated that "we open source the environment and make it easy to install and use". I think acceptance should definitely be conditional on this release given the testbed itself is the central contribution.
> >
> > Thank you for raising this point. The Python package with environments is already available and ready to use, but the link is not included in the paper to preserve anonymity. We have updated our supplementary material file to include an anonymized version of the code.
> >
> > > For the MSE shown in Figure 4, it would be good to give some reference to understand the magnitude of these values relative to the maze (is each maze cell one unit?). Additionally, it would be good to show root MSE, or simply mean distance, instead of MSE so that one can more easily relate these errors to distances in the maze.
> >
> > Thank you for the suggestion, we agree that plotting RMSE instead of MSE would be easier to interpret and will update the plots. The distance is indeed measured in units of maze grid cell.
> >
> > > In Introduction (page 1): "solve later solve" / In Introduction (page 2): "this offers and estimate" / Section 5 (page 8): "RSSM were relevant"
> >
> > Thank you for spotting these, we have fixed them in the paper.

---

> > > ### Comment · Reviewer_9HKy · 2022-11-18
> > > **Thank You**
> > >
> > > Thanks for your response, I feel most of my suggestions have been addressed and I maintain my recommendation of acceptance.

---

### Official Review · Reviewer_kp2x · 2022-10-29

**Confidence:** 4
**Correctness:** 4
**Technical Novelty And Significance:** 3
**Empirical Novelty And Significance:** 3
**Recommendation:** 8

**Clarity, Quality, Novelty And Reproducibility:**

### Quality
Overall, the quality is good.

### Clarity
The paper is clearly written.

### Originality
The tasks are not totally new, but it is the authors' orginal work to systematically publish the task set for research use.

### Reproducibility
The authors agree to open source.


**Details Of Ethics Concerns:**

The paper used a human subject for data collection. More explanation may be required (e.g., the subject agrees to provide his/her data for research use)?

**Strength And Weaknesses:**

## [Strength]
- The study addresses an important open question in decision making
- The proposed environments are intuitive while challenging.
- The paper was written excellently. Both the high-level idea and the details are clear.
- The experiments are comprehensive and convincible with abalation studies.

## [Comments]
- The action space seems to be discrete (Table F.1). I wonder whether the authors can provide a version with continous actions since it is more realistic.
- The title "3D maze" is a bit ambiguous: the agent is moving on a 2D space.
- Some references can be more formal, e.g., "world models .. arxiv" should be the NeurIPS version, and "rl" should be capital letters.


**Summary Of The Paper:**

This paper aims to reinforce the research about memory-dependence in decision making, especially reinforcement learning (RL). The core contribution is the introduction of a new task(environment) set, a kind of Maze with first-person image as observation. To get more rewards, the agent needs to "remember" (or "understand" with internal representations) the layout of the maze so as to reach each target position. The authors performed extensive deep RL experiments to verify (1) Long-term memory is the leading difficulty of the tasks (2) The tasks difficulties range from "solvable" to "challenging" to deep RL agents.  Also, the the authors provide human-demonstrated dataset for offline RL studies. In sum, the paper contributes by introducing a novel benchmark for memory-dependent decision-making studies.

**Summary Of The Review:**

Based on the comments above, I think this is a good paper and I vote for acceptance.

---

> ### Author Response · Authors · 2022-11-15
> **Response to Reviewer kp2x**
>
> Thank you for your comments and for appreciating the quality of the environment and the usefulness of the benchmark!
>
> > The action space seems to be discrete (Table F.1). I wonder whether the authors can provide a version with continous actions since it is more realistic.
>
> Thank you for the suggestion, that would indeed be useful, and quite easy to do. We will include it in the next update of the code.
>
> > The title "3D maze" is a bit ambiguous: the agent is moving on a 2D space.
>
> We agree the term is somewhat ambiguous. The agent indeed is moving on a 2D plane but observes the world through a camera as a 3D scene. We wished to emphasize the 3D aspect, to distinguish it from 2D grid world environments. We have included a clarification of this point in the paper.
>
> > Some references can be more formal, e.g., "world models .. arxiv" should be the NeurIPS version, and "rl" should be capital letters.
>
> Thank you for spotting these, we have fixed them in the paper.

---

> > ### Comment · Reviewer_kp2x · 2022-11-18
> > **Thanks for the reply**
> >
> > Thanks for your reply. I keep my recommendation of acceptance.

---

### Decision · Program_Chairs · 2023-01-20

**Decision:**

Accept: poster

**Justification For Why Not Higher Score:**

As explained above, the previous works on benchmarks and environments should be discussed in the paper.

**Justification For Why Not Lower Score:**

N/A

**Metareview: Summary, Strengths And Weaknesses:**

The paper proposes a benchmark for reinforcement learning approaches, and the focus is mainly on long-term memory. The paper evaluates a series of baseline online and offline models.

Strength:
The paper is clearly written, and the benchmark and the models are explained well.

Weaknesses:

The entire body of literature on Embodied AI has been ignored in the paper.

(1) There are various environments with much more complexity in terms of appearance and scene layout such as:
- Habitat, Savva et al., ICCV 2019
- AI2-THOR, Kolve et al., arXiv 2017
- iGibson, Shen et al., IROS 2021
- ProcTHOR, Deike et al., NeurIPS 2022
- HM3D, Ramakrishnan et al., arXiv 2021

The proposed environment is simplistic compared to these environments. These environments can be configured in a way to focus on long-term memory only (for example, by providing groundtruth perception to factor out the noise in perception).

(2) The task proposed for the above environments also focus on long-term memory. For example,
- Object-goal navigation: ObjectNav Revisited: On Evaluation of Embodied Agents Navigating to Objects, Batra et al. 2020.

- Point-goal navigation (which factors out exploration): Wijmans et al., DD-PPO: Learning Near-Perfect PointGoal Navigators from 2.5 Billion Frames, ICLR 2020.

- Multi-object navigation: Wani et al., MultiON: Benchmarking Semantic Map Memory using Multi-Object Navigation, NeurIPS 2020.

(3) There are also challenges held for these tasks. For example,
- https://aihabitat.org/challenge/2022/
- https://ai2thor.allenai.org/robothor/cvpr-2021-challenge/

The AC follows the recommendation of the reviewers and accepts the paper. However, the authors should add a discussion of the above works to the paper.

**Note From Pc:**

if the above contains the word "oral" or "spotlight" please see: "oral" presentation means -> notable-top-5% and "spotlight" means -> notable-top-25%. As stated in our emails, we are disassociating presentation type from AC recommendations